# Single and Combined Impact of Semaglutide, Tirzepatide, and Metformin on β-Cell Maintenance and Function Under High-Glucose–High-Lipid Conditions: A Comparative Study

**DOI:** 10.3390/ijms26010421

**Published:** 2025-01-06

**Authors:** Esmaeel Ghasemi Gojani, Bo Wang, Dongping Li, Olga Kovalchuk, Igor Kovalchuk

**Affiliations:** Department of Biological Sciences, University of Lethbridge, Lethbridge, AB T1K 3M4, Canada; esmaeel.ghasemigojan@uleth.ca (E.G.G.); bo.wang5@uleth.ca (B.W.); dongping.li@uleth.ca (D.L.); olga.kovalchuk@uleth.ca (O.K.)

**Keywords:** β-cells, HG-HL conditions, metformin, semaglutide, tirzepatide

## Abstract

Type 2 diabetes (T2D), the most common form, is marked by insulin resistance and β-cell failure. β-cell dysfunction under high-glucose–high-lipid (HG-HL) conditions is a key contributor to the progression of T2D. This study evaluates the comparative effects of 10 nM semaglutide, 10 nM tirzepatide, and 1 mM metformin, both alone and in combination, on INS-1 β-cell maintenance and function under HG-HL conditions. INS-1 cells were pretreated for 2 h with single doses of metformin (1 mM), semaglutide (10 nM), tirzepatide (10 nM), or combinations of 1 mM metformin with either 10 nM semaglutide or 10 nM tirzepatide, followed by 48 h of HG-HL stimulation. The results indicate that combining 1 mM metformin with either 10 nM semaglutide or 10 nM tirzepatide significantly enhances the effects of 10 nM semaglutide and 10 nM tirzepatide on HG-HL-induced apoptosis and dysregulated cell cycle. Specifically, the combination treatments demonstrated superior restoration of glucose-stimulated insulin secretion (GSIS) functionality compared to 1 mM metformin, 10 nM semaglutide, and 10 nM tirzepatide.

## 1. Introduction

Type 2 diabetes (T2D) is a chronic metabolic disorder characterized by insulin resistance and impaired insulin secretion, resulting in sustained hyperglycemia. Its global prevalence has reached alarming levels, driven by aging populations, sedentary lifestyles, and unhealthy diets. According to the 2023 International Diabetes Federation (IDF) Diabetes Atlas, an estimated 537 million adults aged 20–79 were living with diabetes globally in 2021, a figure projected to rise to 643 million by 2030 and 783 million by 2045. T2D accounts for over 90% of all diabetes cases, making it the most widespread form of the disease [1].

T2D is a complex metabolic disorder characterized by chronic hyperglycemia resulting from insulin resistance and progressive pancreatic β-cell dysfunction. The core issue in T2D is the reduced ability of pancreatic β-cells to function properly. This problem develops when the cells are exposed to high levels of sugar (glucotoxicity) and fat (lipotoxicity) for extended periods. These metabolic stressors contribute to β-cell apoptosis, endoplasmic reticulum (ER) stress, and diminished insulin production [2,3,4,5].

β-cells exposed to high-glucose–high lipid (HG-HL) stress exhibit significant reductions in glucose-stimulated insulin secretion (GSIS) due to impaired mitochondrial function and altered expression of key β-cell transcription factors, such as Pdx-1, Foxo1, NeuroD1, Nkx6.1, and MafA. These changes, along with epigenetic modifications, collectively contribute to β-cell dedifferentiation, impairing insulin secretion and promoting dysfunction [6,7].

Addressing both insulin sensitivity and β-cell preservation is vital for managing T2D. Metformin, a first-line therapy, enhances insulin sensitivity by activating AMP-activated protein kinase (AMPK), reducing hepatic gluconeogenesis, and mitigating oxidative stress [8,9,10,11,12]. It has been extensively studied for its role in modulating β-cell apoptosis in HG-HL environments, as well as for improving β-cell function by enhancing GSIS and potassium-stimulated insulin secretion (KSIS) [13,14,15,16,17,18,19].

The development of GLP-1 receptor agonists (GLP-1RAs), such as semaglutide, has significantly advanced the management of T2D. Designed with optimal albumin binding for prolonged action, semaglutide enhances glycemic control, supports weight loss, improves cardiovascular outcomes, and preserves β-cell function. It works through GLP-1 receptors in the pancreas and brain, influencing insulin secretion and appetite regulation. Beyond T2D, semaglutide is under investigation for treating obesity and non-alcoholic fatty liver disease [20]. The impact of semaglutide on β-cells involves improving GSIS, reducing apoptosis, and ameliorating ER stress by enhancing incretin signaling and regulating intracellular calcium dynamics [21].

Tirzepatide (LY3298176) is a weekly injectable medication that activates both GIP and GLP-1Rs. It is modified with a fatty acid to bind to albumin, which extends its effectiveness and protects it from being broken down by the DPP4 enzyme. Tirzepatide primarily targets the GIP receptor more strongly than the GLP-1R. While it has lower potency and less activation of the GLP-1 receptor compared to natural GLP-1, it still provides effective blood sugar and weight management by fully activating the GIP receptor. Tirzepatide has shown superior efficacy compared to GLP-1RAs alone, as it simultaneously enhances insulin secretion, suppresses glucagon release, and reduces lipotoxicity [21]. Tirzepatide has been found to improve markers of β-cell function, such as increased HOMA2-B indices and reduced proinsulin levels, proinsulin/C-peptide ratios, and proinsulin/insulin ratios, indicating better insulin processing and reduced β-cell stress [22,23]. In this regard, it has been found that tirzepatide significantly increased the expression of insulin-related genes (*Ins1*, *Ins2*, *Glut2*), β-cell function genes (*Pdx-1*, *MafA*, *NeuroD*), and genes associated with insulin exocytosis (*Munc18*) compared to the control group, with higher expression levels than semaglutide [24].

Although the combinatory impact of metformin with semaglutide and tirzepatide has been studied in the context of the treatment of T2D and weight loss [25,26,27,28,29], there is still a need to elucidate the combined impact of metformin with any of these medications on the underlying mechanisms in β-cells and other target cells.

This study investigates the effects of metformin, semaglutide, tirzepatide, and their combinations—metformin with semaglutide (MS) and metformin with tirzepatide (MT)—on β-cell health under HG-HL conditions. It examines critical factors such as apoptosis, cell cycle dysregulation (specifically S-phase entry), GSIS, insulin production, and the expression of β-cell-specific genes. This study aims to uncover the mechanisms driving the effects of these therapies and evaluate the enhanced benefits of combination treatments in preserving β-cell function and maintenance under HG-HL stress, providing valuable insights into innovative strategies for improving T2D management.

## 2. Results

### 2.1. 1 mM Metformin and Its Combinations with 10 nM Semaglutide and 10 nM Tirzepatide Exhibit Enhanced Modulatory Effects on HG-HL-Induced Apoptosis and Apoptotic Biomarkers

All three medications at the defined doses used in this experiment, as well as the combinations of 1 mM metformin with both 10 nM semaglutide and 10 nM tirzepatide, mitigate HG-HL-induced apoptosis in INS-1 β-cells (Figure 1A). Although 1 mM metformin demonstrates a greater modulatory effect on HG-HL-induced apoptosis than either 10 nM semaglutide or 10 nM tirzepatide, the combinations of semaglutide or tirzepatide with metformin achieve a similar level of apoptosis mitigation as 1 mM metformin, suggesting the absence of any potentiation or additive effects. The modulatory impact of both 10 nM semaglutide and 10 nM tirzepatide on HG-HL-induced apoptosis was comparable (Figure 1A). Consistently, Western blot analysis of apoptotic biomarkers, including Cleaved PARP (C-PARP), Cleaved Caspase-3 (C-Caspase-3), Cleaved Caspase-9 (C-Caspase-9), Bax, Bim_EL_, and Bim_L_, supports the results of the apoptosis assay (Figure 1B). Interestingly, treatment with 10 nM tirzepatide increases the expression of certain apoptotic markers, such as C-Caspase-3, C-Caspase-9, Bax, and Bim_EL_ (Figure 1B(c,d,f)). In most cases, 1 mM metformin combination with 10 nM semaglutide performed better than with 10 nM tirzepatide. Western blot analysis of C-Caspase-3 reveals that the MS and MT treatments exert a greater modulatory effect on its levels compared to treatments with 1 mM metformin, 10 nM semaglutide, or 10 nM tirzepatide (Figure 1B(f)).

### 2.2. 1 mM Metformin and Its Combinations with Both 10 nM Semaglutide and 10 nM Tirzepatide Modulate HG-HL-Induced Cell Cycle Dysregulation

Compared to the control, HG-HL conditions significantly increase S-phase entry and reduce G2-phase entry in INS-1 β-cells, likely due to the stress imposed by HG-HL levels. Treatments with 1 mM metformin, as well as in combination with both 10 nM semaglutide and 10 nM tirzepatide, modulate the HG-HL-induced entry into the S-phase. Although 1 mM metformin and the MS combination show similar modulatory effects on induced S-phase entry, the MT combination demonstrates a slightly less pronounced effect than 1 mM metformin. Notably, 10 nM semaglutide and 10 nM tirzepatide actually increase S-phase entry under HG-HL conditions, indicating a distinct effect from metformin. In contrast, 1 mM metformin, MS, and MT treatments increase the proportion of cells entering the G2-phase. The 10 nM semaglutide has no notable effect on G2-phase entry, while 10 nM tirzepatide reduces G2-phase entry under HG-HL conditions (Figure 2).

### 2.3. 1 mM Metformin and Its Combinations with Both 10 nM Semaglutide and 10 nM Tirzepatide Improve HG-HL-Reduced GSIS in INS-1 β-Cells

HG-HL induction impairs the secretory function of β-cells, affecting basal glucose levels, KSIS, and GSIS. Treatment with 1 mM metformin improved all three parameters in HG-HL-induced INS-1 β-cells. However, both 10 nM semaglutide and 10 nM tirzepatide, at the specific doses used in this experiment, had a negative impact on impaired GSIS (Figure 3). While 10 nM semaglutide, 10 nM tirzepatide, and their combinations with metformin had no significant effect on basal glucose levels and KSIS, combining 1 mM metformin with either 10 nM semaglutide or 10 nM tirzepatide significantly enhanced GSIS in HG-HL-induced INS-1 β-cells. Notably, these combined treatments yielded a greater improvement in GSIS than 1 mM metformin (Figure 3a,b). Although all treatments positively impacted total insulin production in HG-HL-induced INS-1 β-cells, 1 mM metformin showed the greatest improvement compared to other treatments, including 10 nM semaglutide, 10 nM tirzepatide, MS, and MT. The 10 nM semaglutide had a slightly better effect on total insulin production than 10 nM tirzepatide (Figure 3c). The qRT-PCR results of insulin genes confirmed that 1 mM metformin, as well as the combinations of MS and MT, were the most effective treatments in upregulating insulin gene expression. In contrast, neither 10 nM semaglutide nor 10 nM tirzepatide showed any restorative effect on *Ins* mRNA levels (Figure 3d,e).

### 2.4. All Treatments Significantly Reduce TXNIP and P-STAT1 Levels in HG-HL-Induced Cells

The Western blotting results show that all treatments reduced the elevated levels of TXNIP and P-STAT1 in HG-HL-induced INS-1 β-cells. Although there were no significant differences between 1 mM metformin and 10 nM semaglutide in reducing TXNIP levels, 10 nM tirzepatide resulted in a lower inhibitory effect on the HG-HL-induced TXNIP expression. Compared to 1 mM metformin and 10 nM semaglutide treatments, the combination of 1 mM metformin and 10 nM semaglutide (MS) demonstrated a more effective modulatory impact (Figure 4).

### 2.5. 1 mM Metformin and MS Restore the Suppressed Expression of Genes Crucial for INS-1 β-Cell Maintenance and Identity Under HG-HL Conditions

Based on our results with 10 nM semaglutide and MS on HG-HL-induced apoptosis and S-phase entry, this part of the experiment was only conducted with 1 mM metformin, 10 nM semaglutide, and MS. Experiments were performed under two conditions: co-treatment and post-treatment. In the co-treatment setup, 1 mM metformin prevented the change in the expression of all studied genes in HG-HL-induced INS-1 β-cells, including *Ins1*, *Ins2*, *Pdx1*, *Slc2a2*, *NeuroD1*, *MafA*, and *Foxo1* (Figure 5A), along with the protein levels of PDX-1 and FOXO1. However, the level of NKX6.1 protein, which was significantly reduced under HG-HL-induced conditions, could not be restored by 1 mM metformin or any other treatment (Figure 5A(b)). The MS treatment showed similar restorative effects to 1 mM metformin on most genes, including *Ins1*, *Ins2*, *Slc2a2*, *Pdx1*, and *MafA*, but had additional outcomes (Figure 5A(a)). MS reduced the expression of NeuroD1 more significantly than in HG-HL-induced β-cells and failed to restore *Foxo1* mRNA or the protein levels of PDX-1 and FOXO1 (Figure 5B). The 10 nM semaglutide treatment restored the expression of only *Ins2* and *Pdx1* genes, and these effects were weaker than those observed with 1 mM metformin (Figure 5A(a)).

In the post-treatment experiment, we observed notable differences from the co-treatment findings. Specifically, 10 nM semaglutide demonstrated a stronger restorative effect on the reduced levels of *Ins1*, *Ins2*, and *Slc2a2* mRNA compared to both 1 mM metformin and MS. Interestingly, 10 mM metformin and its combination with 10 nM semaglutide were less effective at restoring the levels of these genes in HG-HL-induced β-cells than 1 mM metformin. In contrast to the above-mentioned genes, all treatments positively influenced the reduced levels of MafA, with all groups showing successful restoration of its expression in HG-HL-induced β-cells. Interestingly, no significant differences were observed in the qPCR results of *Pdx-1*, *Foxo1*, and *NeuroD1* across all treatments (Figure 5B(a)). Under post-treatment conditions, the HG-HL+Med group (exposed to HG-HL conditions for 48 h, followed by 48 h of incubation in a normal medium) showed the highest NKX6.1 protein levels, surpassing even the control group (Ct) (Figure 5B(b)). The 10 nM semaglutide positively affected NKX6.1 levels, though less so than the HG-HL+Med group but significantly higher than the Ct group. In contrast, 1 mM metformin and the MS combination had similar effects on NKX6.1 protein levels, which were comparable to the Ct group and lower than both HG-HL+Med and 10 nM semaglutide treatments (Figure 5B(b)).

## 3. Discussion

Obesity contributes to the dysfunction and reduction in the number of pancreatic β-cells through glucolipotoxicity. In obese individuals, elevated levels of free fatty acids and lipopolysaccharides negatively impact β-cell functionality and survival through mechanisms such as ER stress, the release of proinflammatory cytokines, and impaired insulin signaling [30]. Chronic exposure to HG-HL is a key factor in β-cell apoptosis [31,32].

In this research, we studied the impact of three antidiabetic medications (1 mM metformin, 10 nM semaglutide, and 10 nM tirzepatide) and a certain combination of these medications on the maintenance and function of INS-1 β-cells stimulated by HG-HL conditions.

### 3.1. The Impact of the Treatments on HG-HL-Induced Apoptosis

Although all five treatments modulated HG-HL-induced apoptosis in INS-1 β-cells, 1 mM metformin, MS, and MT exhibited greater anti-apoptotic effects compared to 10 nM semaglutide and 10 nM tirzepatide (Figure 1A). The anti-apoptotic properties of 1 mM metformin are well documented [33,34,35,36,37], demonstrating its ability to reduce apoptosis through various mechanisms. These include the activation of AMPK and the inhibition of mTOR signaling [38], the suppression of CD36 expression [16], the activation of NF-κB signaling [39], the reduction in TXNIP levels [39,40], and the mitigation of oxidative stress [41,42]. These effects are associated with metformin’s ability to preserve mitochondrial function, lower the Bax/Bcl-2 ratio, and inhibit Caspase-3 activation, collectively contributing to its protective role in promoting cell survival [33]. Regarding the role of the modulatory impact of metformin and TXNIP, it has been found that TXNIP activation, induced by HG, leads to oxidative stress and inflammation through ROS production and NLRP3 inflammasome activation. Metformin treatment mitigates these effects by promoting TXNIP degradation, enhancing thioredoxin (Trx) activity, reducing ROS levels, and inhibiting NLRP3-mediated inflammation [43].

Consistent with these findings, our study suggests that 1 mM metformin reduces apoptosis, which may be related to downregulating TXNIP (Figure 4), leading to decreased ROS levels and subsequent reductions in apoptotic biomarkers, such as Bax, Bim, C-Caspase 9, C-Caspase-3 proteins, and C-PARP, and thereby apoptosis (Figure 1B).

Earlier studies suggested that TXNIP transcription regulation may follow a tissue-specific pattern. For example, the transcription factor FOXO1 promotes TXNIP expression in neurons and endothelial cells but suppresses it in the liver and pancreatic β-cells [44]. FOXO1 is a key regulator of TXNIP expression in pancreatic beta cells, where it binds to the TXNIP promoter and inhibits its transcription. This inhibition occurs through a competitive interaction with ChREBP, a transcription factor responsible for glucose-induced TXNIP expression. FOXO1 prevents ChREBP from binding to the TXNIP promoter, without affecting ChREBP expression or localization. Additionally, FOXO1 also inhibits transcription of other ChREBP target genes, such as liver pyruvate kinase [45]. This aligns with our findings that 1 mM metformin treatment increases the level of FOXO1 in HG-HL-induced β-cells, which, in turn, mitigates the levels of TXNIP, ROS, apoptotic biomarkers, and apoptosis (Figure 5A(b)).

We observed that similar to 1 mM metformin, 10 nM semaglutide effectively mitigated HG-HL-induced apoptosis in INS-1 β-cells. However, its modulatory effects on apoptosis and apoptotic biomarkers, including Bax, C-caspase-3, C-Caspase-9, C-PARP, and Bim proteins, were less pronounced compared to metformin (Figure 1A and Figure 2a). Consistently, previous studies have demonstrated that semaglutide, along with liraglutide (another GLP-1R agonist), reduces apoptosis in diabetic contexts [46,47]. This protective effect is attributed to its cytoprotective actions through cAMP-dependent survival pathways [48], its ability to enhance GSIS, and its role in reducing oxidative stress [49,50]. In alignment with these findings, our study revealed that 10 nM semaglutide mitigates HG-HL-induced TXNIP expression (Figure 4), which in turn reduces ROS levels. These results provide further evidence that TXNIP may serve as an upstream target through which semaglutide exerts its modulatory effects on HG-HL-induced apoptosis. Unlike 1 mM metformin, the modulatory impact of 10 nM semaglutide on HG-HL-induced TXNIP is not mediated through increasing FOXO1 protein.

Our results indicate that similar to 1 mM metformin and 10 nM semaglutide, 10 nM tirzepatide modulated HG-HL-induced apoptosis and the apoptotic biomarkers BimL and C-PARP in INS-1 β-cells (Figure 1A), which may be mediated through its impact on HG-HL-induced TXNIP production (Figure 4). However, unlike 1 mM metformin and 10 nM semaglutide, 10 nM tirzepatide also increased the levels of additional apoptotic biomarkers such as C-Caspase-3, C-Caspase-9, and Bim EL (Figure 1B). The response pattern of 10 nM tirzepatide on these biomarkers closely aligns with the outcomes of the apoptosis assay (Figure 1B) and the behavior of TXNIP protein (Figure 4) under HG-HL conditions. This observation may be linked to prolonged exposure to GIP or GIPR agonists, which have been shown to lead to receptor desensitization, reducing their effectiveness. Studies indicate that extended GIPR activation can result in receptor desensitization, potentially explaining the stimulatory effects of tirzepatide on some apoptotic biomarkers [51,52,53]. In this regard, it has been observed that similar to the remarkable therapeutic effectiveness of dual GIPR/GLP-1R agonists, the combination of GLP-1R agonism with GIPR antagonism also demonstrates significant therapeutic benefits [53,54,55]. These findings underscore the complexity of GIPR mechanisms of action and the synergistic interplay between GIPR and GLP-1R, emphasizing the need for further research to fully elucidate their dual metabolic and apoptotic effects.

One of the main goals of this study was to investigate the combined effects of metformin with semaglutide and tirzepatide on the maintenance and function of β-cells under HG-HL conditions. Our results show that neither MS nor MT had a greater effect on HG-HL-induced apoptosis compared to 1 mM metformin. However, the combined treatments had a stronger modulatory effect on apoptosis than either 10 nM semaglutide or 10 nM tirzepatide. This suggests that combining 10 nM tirzepatide and 10 nM semaglutide with 1 mM metformin enhances their ability to modulate apoptosis in the context of T2D.

C-PARP is a key marker and mediator of apoptosis. During apoptosis, Caspase-3 cleaves full-length PARP-1 into 89 kDa and 24 kDa fragments, inactivating PARP’s DNA repair function and preventing NAD+ and ATP depletion. The 24 kDa fragment may inhibit uncleaved PARP, while the 89 kDa fragment contributes to apoptosis despite reduced catalytic activity. PARP cleavage occurs downstream of caspase activation in both the extrinsic and intrinsic apoptosis pathways [56,57,58].

Based on our results (Figure 1B), since all treatments modulated the levels of C-PARP in HG-HL-induced INS-1 β-cells and the response pattern of C-PARP was similar to other apoptotic biomarkers, we suggest that PARP cleavage acts as a final target for all treatments in their regulation of HG-HL-induced apoptosis.

### 3.2. The Impact of the Treatments on HG-HL Cell Cycle Dysregulation

HG-HL conditions induce stress that triggers cellular adaptations to enhance survival, including a shift in the cell cycle that increases the proportion of cells in the S-phase to support DNA replication necessary for proliferation or survival responses. Glucolipotoxicity is known to generate ROS, cause ER stress, and disrupt mitochondrial function, leading to impaired cellular signaling and the dysregulation of normal cell cycle checkpoints [59]. This dysregulation could be associated with triggering compensatory adaptations for survival. These adaptations often include promoting entry into the S-phase (DNA synthesis phase) and reducing entry into the G2-phase, likely due to the inability to move past DNA replication (Figure 2).

Our results indicate that treatments with 1 mM metformin and MS or MT modulate HG-HL-induced metabolic stress and partially alleviate the dysregulation of the cell cycle in the S-phase and G2-phase (Figure 2). Specifically, 1 mM metformin in combination with both 10 nM semaglutide and 10 nM tirzepatide, appears to mitigate the HG-HL-induced dysregulation of the cell cycle by reducing excessive S-phase entry and restoring G2-phase entry. Interestingly, 10 nM semaglutide and 10 nM tirzepatide tend to exacerbate stress, promoting increased S-phase entry and decreased G2-phase entry. However, when combined with 1 mM metformin, these effects are alleviated (Figure 2), suggesting that the combination of these medications with metformin may help reduce their potential adverse impact on cell cycle regulation.

Since the patterns of S-phase and G2-phase entry responses to the treatments align with the patterns of C-PARP responses, it is likely that C-PARP activity plays a crucial role in mediating the treatments’ effects on cell cycle dysregulation. In this context, it has been observed that the inhibition of PARP activity through its cleavage can destabilize replication forks, leading to replication stress and potentially excessive entry into the S-phase as cells attempt to compensate for incomplete DNA synthesis [60].

Based on our results and considering that the pattern of HG-HL-induced TXNIP responses to the treatments aligns with C-PARP responses, we propose that TXNIP may function as an upstream target for the treatments, mediating their effects on C-PARP and, consequently, on the cell cycle (Figure 4). The treatments likely exert their effects through the modulation of TXNIP, ROS, ER stress, and inflammatory responses.

### 3.3. The Impact of the Treatments on HG-HL-Impaired GSIS

To study the impact of treatments on β-cell function, we examined the responses of GSIS, KSIS, total insulin production, and expression of insulin-coding genes in HG-HL-induced INS-1 β-cells. As our results indicate, 1 mM metformin, MS, and MT all improved the reduced GSIS levels in HG-HL-induced INS-1 β-cells (Figure 3). These findings align with previous studies, which demonstrated that metformin significantly enhances impaired GSIS in INS-1β-cells [17,61]. The protective effects of metformin on GSIS are linked to its activation of the AMPK pathway ^17^. Interestingly, the combined treatments of MS and MT were more effective than 1 mM metformin in improving GSIS, suggesting that these combinations might offer more potent therapeutic options for impaired GSIS in HG-HL-induced INS-1 β-cells (Figure 4). In addition to GSIS, HG-HL conditions can impair the basal secretory ability of INS-1 β-cells. Remarkably, we observed that 1 mM metformin could, at least in part, restore this function (Figure 3). This restoration may be attributed to the effects of metformin on total insulin expression and production, as well as its ability to inhibit HG-HL-induced apoptosis in INS-1 β-cells. Neither 10 nM semaglutide nor 10 nM tirzepatide improved the impaired level of GSIS under HG-HL stress in our model.

A previous study indicates that semaglutide significantly enhances GSIS in individuals with T2D, improving both first- and second-phase insulin secretion. It also increases insulin secretion rates to levels similar to those in healthy participants and boosts maximal insulin secretion capacity [62,63]. Using an obese T2D mouse model, it has been found that while no significant difference in GSIS was observed between the control and semaglutide groups, GSIS was significantly greater in the tirzepatide group compared to the control group [24]. In contrast to these results, we found that both 10 nM semaglutide and 10 nM tirzepatide had a negative impact on the impaired GSIS in HG-HL-induced β-cells. It is worth mentioning that the previous findings were derived from in vivo studies [29], which suggests that the effects of these two medications may not be directly related to their impact on β-cells. For instance, both semaglutide and tirzepatide have well-established effects on weight loss [64,65,66,67,68]. It is possible that the beneficial effects of these medications on GSIS in in vivo systems result from their ability to reduce obesity, which in turn improves impaired GSIS in HG-HL-induced β-cells. In other studies, where the positive impact of semaglutide and tirzepatide was observed, the effects of higher doses on GSIS were assessed in uninduced INS-1 β-cells [63,69,70]. However, in our study, we investigated the effect of a single dose of semaglutide and tirzepatide (10 nM) on impaired GSIS in HG-HL-induced INS-1 β-cells. The concentration we selected was based on pharmacokinetic data from clinical trials, as it approximates the therapeutic drug levels observed in human plasma [71,72] while also considering the combination of these treatments with metformin.

Finally, we also assessed the response of KSIS to all treatments, adding another layer to our understanding of how metformin and its combinations impact β-cell function under HG-HL-induced stress.

The primary difference between GSIS and KSIS lies in their mechanisms. GSIS follows the full physiological insulin secretion pathway, including glucose metabolism, ATP/ADP ratio increase, closure of ATP-sensitive potassium channels, membrane depolarization, calcium influx, and insulin granule exocytosis. In contrast, KSIS bypasses glucose metabolism, directly triggering insulin release through cell membrane depolarization. KSIS isolates late-stage mechanisms, aiding in assessing secretory function and pinpointing defects in glucose metabolism or sensing [73,74,75,76]. Comparing GSIS and KSIS provides insights into β-cell function, highlighting specific disruptions in stimulus–secretion coupling under physiological and pathological conditions. Our results show that only 1 mM metformin treatment significantly increased the reduced KSIS levels in HG-HL-induced β-cells. While both MS and MT also elevated this response, the effect was not statistically significant. This suggests that 1 mM metformin treatment improves the secretory ability of HG-HL-induced INS-1 β-cells. However, when combined with either 10 nM semaglutide or 10 nM tirzepatide, this effect was neutralized (Figure 3b). The positive impact of metformin on impaired KSIS has been supported by other studies. Accordingly, it has been found that metformin helps preserve pancreatic β-cell function and maintain insulin secretion under certain conditions, possibly through its effects on AQP7 expression and the MAPK signaling pathway [18,77].

In line with previous studies [78], at least part of the modulatory effects of 1 mM metformin, MS, and MT on GSIS can be explained by their stimulatory effects on total insulin expression and production in HG-HL-induced β-cells (Figure 3). This is further supported by the observation that the patterns of 1 mM metformin’s effects on total insulin content and GSIS are similar.

### 3.4. The Impact of the Treatments on the Response of β-enriched Genes and Proteins

In the second part of this study, we investigated the responses of genes and proteins associated with β-cell identity, including *Ins1*, *Ins2*, *Pdx1*, *Slc2a2*, *MafA*, *NeuroD1*, and *Foxo1* mRNA, as well as PDX-1, FOXO1, and NKX6.1 proteins, to 1 mM metformin, 10 nM semaglutide, and the combination of 1 mM metformin and 10 nM semaglutide (MS). Given the superior effects of 10 nM semaglutide and MS compared to 10 nM tirzepatide and MT observed in earlier experiments, we focused on 1 mM metformin, 10 nM semaglutide, and MS for these analyses.

Experiments were performed under two conditions: co-treatment and post-treatment. In the co-treatment setup, 1 mM metformin restored the expression of all studied genes in HG-HL-induced β-cells, including *Ins1*, *Ins2*, *Pdx1*, *Slc2a2*, *MafA*, *NeuroD1*, and *Foxo1*, along with the protein levels of PDX-1 and FOXO1. This suggests that 1 mM metformin may play a critical role in preserving β-cell identity by counteracting HG-HL-induced dedifferentiation. In line with our results, it has been found that metformin inhibits islet β-cell dedifferentiation by decreasing C3, Nga3, and Oct4 while increasing Pdx1 and MafA expression, with C3 promoting dedifferentiation through the Wnt/β-catenin pathway [79].

However, the level of NKX6.1 protein, which was significantly reduced under HG-HL-induced conditions, could not be restored by 1 mM metformin or any other treatment, suggesting the complex role of the Nkx6.1 gene in the maintenance and function of β-cells.

The MS combination treatment had similar effects to 1 mM metformin on most genes, such as *Ins1*, *Ins2*, *Slc2a2*, *Pdx1*, and *MafA*, but also showed additional results. MS more significantly reduced the expression of NeuroD1 compared to HG-HL-induced β-cells and did not restore Foxo1 mRNA or the protein levels of PDX-1 and FOXO1 (Figure 5A). These findings suggest that MS is not superior to 1 mM metformin in maintaining β-cell identity.

The impact of GLP-1 on the mitigation of β-cell dedifferentiation is well established [7]. Accordingly, we found that 10 nM semaglutide treatment partially restored the expression of *Ins2* and *Pdx1* genes, with these effects being less pronounced than those seen with 1 mM metformin. This suggests that 10 nM semaglutide is less effective than 1 mM metformin in maintaining β-cell identity under HG-HL-induced conditions (Figure 5A). However, combining 10 nM semaglutide with 1 mM metformin partially enhanced its effect, suggesting a greater potential for improving β-cell identity maintenance.

This analysis highlights that 1 mM metformin remains the most effective treatment among those tested for counteracting β-cell dedifferentiation and preserving β-cell identity under HG-HL stress conditions.

In the post-treatment experiment, we observed notable differences from the co-treatment findings. Specifically, 10 nM semaglutide demonstrated a stronger restorative effect on the reduced levels of *Ins1*, *Ins2*, and *Slc2a2* mRNA compared to both 1 mM metformin and MS. Interestingly, 1 mM metformin and its combination with 10 nM semaglutide (MS) were less effective at restoring the levels of these genes in HG-HL-induced β-cells. This suggests that the protective effect of 1 mM metformin is stress dependent and that, in the absence of stress, it may have adverse effects on the normal function of β-cells.

In contrast to the above genes, all treatments positively influenced the reduced levels of MafA, with all groups showing successful restoration of its expression in HG-HL-induced β-cells.

The most intriguing result was observed at the protein level with NKX6.1. Under post-treatment conditions, the HG-HL+Med group exhibited the highest levels of NKX6.1 protein, surpassing even the control group (Ct). While 10 nM semaglutide showed a positive effect on NKX6.1 levels, its impact was lower than the HG-HL+Med group but significantly higher than the Ct group. Conversely, the effects of 1 mM metformin and MS on NKX6.1 protein levels were comparable to the Ct group and notably lower than the HG-HL+Med- and 10 nM semaglutide-treated groups.

These results further suggest that NKX6.1 is regulated by a complex mechanism at the cellular level. This highlights the need for additional research to better understand the specific role and function of this protein in β-cells.

## 4. Materials and Methods

### 4.1. Chemicals and Reagents

Semaglutide (CAS No. 910463-68-2) and tirzepatide (CAS No. 2023788-19-2) were from Adipogen Corp. (San Diego, CA, USA). Metformin hydrochloride (CAS No. 1115-70-4), L-Glutamine (TMS-002-C), sodium pyruvate (S8636), HEPES (TMS-003-C), β-mercaptoethanol (ES-007), and INS-1 832/13 Rat Insulinoma Cell (SCC207) were purchased from EMD Millipore Corporation (Temecula, CA, USA). Roswell Park Memorial Institute Medium (RPMI-1640) (350-060-CL) and D-PBS, 1X(311-010-CL) were acquired from Wisent Inc. (Saint-Jean-Baptiste, QC, Canada).

### 4.2. Cell Culture and Treatments 

INS-1 832/13 Rat Insulinoma Cells were cultured in RPMI 1640 medium supplemented with 2 mM L-Glutamine, 1 mM sodium pyruvate, 10 mM HEPES, 0.05 mM β-mercaptoethanol, and 10% fetal bovine serum (FBS) ((10082147) acquired from Fisher Scientific Company (Ottawa, ON, Canada), along with 2.5 mM glucose. Cells within passages 5-10 were used for all experiments. After being cultured in a medium containing 2.5 mM glucose for two days, cells were exposed to a specified dosage of individual treatments, their combinations, and/or an equal quantity of the vehicle for a duration of 2 h. All three treatments used in this study were dissolved in D-PBS. We used a single dose of each treatment: 1 mM metformin [37,80], 10 nM semaglutide, and 10 nM tirzepatide [81,82,83]. The concentrations of semaglutide and tirzepatide commonly used in in vitro studies typically range from 10 nM to 100 nM. These concentrations are selected to approximate therapeutic doses while minimizing potential cytotoxic effects [63,69,70,81,82,83,84,85]. Subsequently, high-glucose (HG) (25 mM glucose) and high-lipid (HL) (400 μM palmitic acid) conditions were imposed for the next 48 h [30]. Following the experimental treatments, the cells were used for various assays, including MTT, apoptosis, cell cycles, GSIS, KSIS, Western blot, and q-RT-PCR. For the post-treatment experiment, cells were first exposed to HG-HL conditions for 48 h. Afterward, the HG-HL medium was replaced with normal culture media containing the specified doses of each treatment, and the cells were incubated for an additional 48 h. This approach allows for the evaluation of the treatments’ effects on β-cell function and dedifferentiation after the stress condition is removed, simulating a recovery or therapeutic window following stress-induced damage.

### 4.3. Apoptosis Assay

The BD Pharmingen™ FITC Annexin V Apoptosis Detection Kit II (Cat No. BDB556570; BD Biosciences, Mississauga, ON, Canada) was used to quantify apoptotic cells by detecting phosphatidylserine (PS) translocation to the cell membrane’s outer leaflet. FITC Annexin V binds to exposed PS on apoptotic cells, while propidium iodide (PI) distinguishes viable cells from non-viable ones. Cells positive for FITC Annexin V alone were classified as apoptotic, propidium iodide (PI)-positive cells as necrotic or late-apoptotic, and cells negative for both as viable. Cells were washed twice with cold D-PBS and then resuspended in 1X Annexin V Binding Buffer at a concentration of 1 × 10^6^ cells/mL. A 100 µL aliquot (1 × 10^5^ cells) was transferred to a 5 mL culture tube, and 5 µL of FITC Annexin V and 5 µL of PI were added. The cell suspension was gently vortexed and incubated in the dark at room temperature (25 °C) for 15 min. After incubation, 400 µL of 1X Binding Buffer was added, and the samples were analyzed by flow cytometry within one hour using a BD Biosciences (Mississauga, ON, Canada) *BD FACSAria™* Fusion Flow Cytometer.

### 4.4. Cell Cycle Assay

Cells were harvested and washed in PBS and then fixed by adding cold 70% ethanol dropwise to the pellet while vortexing to prevent clumping, followed by a 30 min fixation at 4 °C. After fixation, cells were washed twice in PBS by centrifuging at 850 g, carefully discarding the supernatant to minimize cell loss. To remove RNA, 50 µL of RNase (100 µg/mL) was added to each sample, followed by 200 µL of PI from a 50 µg/mL stock solution to stain DNA. After incubating for 30 min at room temperature in the dark, PI emission was detected using the BD FACSAria™ Fusion Flow Cytometer (BD Biosciences, Mississauga, ON, Canada). This system allows for high-resolution analysis of cell fluorescence, ensuring precise detection of propidium iodide (PI) signals during flow cytometric analysis.

### 4.5. Western Blotting

The cells were rinsed twice with cold PBS and then lysed using a RIPA buffer. After centrifugation at 13,000 rpm for 15 min, the supernatant was collected and transferred to a new microtube. The protein content was determined using the Bradford assay, and the lysates were used for Western blotting. Proteins were separated on polyacrylamide gels of varying concentrations (8%, 10%, and 12%) and transferred to polyvinylidene difluoride (PVDF) Amersham Hybond^®^ P membranes (RPN2020F) acquired from GE Healthcare, Oakville, ON, Canada. The membranes were blocked with a PBS containing 1% tween 20 (PBST) solution and 5% milk before being incubated with primary antibodies overnight at 4 °C. After three washes with PBST, the membranes were incubated with secondary antibodies for two hours at room temperature and washed again with PBST. Immunoreactivity was identified using peroxidase-conjugated antibodies, and the ECL Plus Western Blotting Detection System (GE Healthcare, Oakville, ON, Canada) was utilized to render the reaction visible. The band intensities were quantified through analysis with ImageJ 1.54 K and normalized relative to the housekeeping protein’s intensity. Primary antibodies used were Cleaved Caspase-3 (Asp175) (Product No. 9664), Caspase 7 (Product No. 12827), Cleaved Caspase-7 (Asp198) (Product No. 8438), Cleaved PARP (Asp214) (Product No. 5625), Bim (Product No. 2933), PDX-1 (Product No. D59H3), FoxO1 (Product No. 2880S), NKX6.1 (Product No. 54551), and TXNIP (Product No. 14716S), which were sourced from Cell Signal Technology (Whitby, ON, Canada). Anti-P-STAT1 (phospho S727) (Cat No. ab109461) and Anti-β Actin (Cat No. ab8227) were purchased from Abcam (Toronto, ON, Canada). The secondary antibodies were obtained from Santa Cruz Biotechnology (Dallas, TX, USA).

### 4.6. qRT-PCR

Total RNA was isolated using TRIzol^TM^ (15596018, Invitrogen, Life Technologies Inc., Burlington, ON, Canada), and its concentration was subsequently determined using a nanodrop instrument (NanoDrop 2000c, ThermoFisher Scientific, Waltham, MA, USA). A portion of the total RNA was used for cDNA synthesis, utilizing 1 μg of total RNA and a cDNA synthesis kit, iScript™ Reverse Transcription Supermix (1708897, BioRad Laboratories, Saint-Laurent, QC, Canada). The resulting cDNA was then used as a template for q-PCR, with 1 μL of cDNA being used per reaction. The qRT-PCR reactions were performed using a SsAdvancedTM Universal Inhibitor-Tolerant SYBR Green Supermix (1725017, Bio-Rad Laboratories (Saint-Laurent, QC, Canada). The primers were designed based on sequences reported in our previous publications [30].

### 4.7. Glucose-Stimulated Insulin Secretion (GSIS) and Potassium-Stimulated Insulin Secretion (KSIS)

GSIS and KSIS assays were conducted to assess β-cell function under HG-HL conditions following treatment. Cells were seeded at a density of 0.5 × 10^4^ per well in a 24-well plate, incubated for 2 days, and then pretreated with specified treatment concentrations 2 h before exposure to HG-HL conditions for 48 h. For GSIS and KSIS assays, cells were washed with HEPES Balanced Salt Solution (HBSS) containing 2.5 mM glucose, followed by a 1 h wash in the same solution. Three wells per treatment were designated for testing: one with HBSS containing 2.5 mM glucose (normal glucose), another with 16.5 mM glucose (high glucose), and a third with 2.5 mM glucose plus 50 mM KCl. After a 2 h incubation, samples were collected for insulin ELISA analysis. The Rat/Mouse Insulin ELISA Kit (EZRMI-13K) from Sigma Aldrich (Oakville, ON, Canada) was used. Each well was washed, loaded with 10 μL of the sample and 80 μL of Detection Antibody, incubated at room temperature for 2 h, washed again, and then treated with 100 μL of Enzyme Solution for 30 min. Wells were washed again, followed by the addition of 100 μL of Substrate Solution, and incubated for 5–20 min before stopping the reaction. Absorbance was measured at 450 nm and 590 nm using a SpectraMax i3x Multi-Mode Microplate Reader (Molecular Devices, San Jose, CA, USA). For the GSIS and KSIS assays, the samples were diluted 150-fold, while for total insulin content, the extracted proteins were diluted 3500-fold to ensure that the insulin levels were within the detectable range of the assay kit.

### 4.8. Statistics

The collected data were analyzed statistically using a one-way analysis of variance (ANOVA) to determine overall significance among groups. Following the ANOVA, Tukey’s post hoc test was performed for multiple comparisons to assess the differences between group means. Statistical analyses were conducted using GraphPad Prism, Version 9.5.1 (GraphPad Software, San Diego, CA, USA). Data are presented as mean ± standard deviation (SD) unless otherwise specified, and a *p*-value of less than 0.05 was considered statistically significant.

## 5. Conclusions

The summary of the current research findings is as follows.

The combination of 1 mM metformin with both 10 nM semaglutide and 10 nM tirzepatide improves the effects of 10 nM semaglutide and 10 nM tirzepatide on cellular apoptosis, dysregulated cell cycle, and impaired function in HG-HL-induced INS-1 β-cells. Although the combinations did not show significant effects on 1 mM metformin’s modulatory impact on apoptosis and cell cycle regulation, they did enhance GSIS in HG-HL-induced INS-1 β-cells.

The effects of 1 mM metformin and other treatments on TXNIP levels in HG-HL-induced INS-1 β-cells mediate their impact on apoptosis and dysregulated cell cycle.

The 1 mM metformin and MS similarly mitigated the HG-HL-induced reduction in key genes associated with β-cell identity, performing better than 10 nM semaglutide and comparable to 1 mM metformin, suggesting that MS could enhance 10 nM semaglutide’s impact on β-cell dedifferentiation.

The post-treatment results differed from the co-treatment outcomes, with 10 nM semaglutide showing a better ability to restore the reduced levels of some β-cell-specific genes, indicating that 1 mM metformin may be more effective in maintaining β-cell function and identity under stress conditions like HG-HL.

## 6. Limitations and Future Directions

While our study provides valuable insights into the effects of the treatments on β-cell maintenance, function, and dedifferentiation under HG-HL conditions, several limitations should be considered. The most significant limitation is the absence of in vivo validation. Although our in vitro findings are promising, an in vivo study would provide a more comprehensive understanding of how these treatments affect β-cell behavior and function in the context of the entire organism. Animal models would help confirm the translational relevance of our results and allow for the evaluation of systemic effects.

Additionally, our study utilized a limited range of doses for each treatment and their combinations. Future studies should explore a broader spectrum of doses to better assess the dose–response relationship and identify the optimal therapeutic window. This would also provide a more detailed understanding of the pharmacological effects at various concentrations, which may vary across different cell types or experimental conditions.

Another area for future research is the inclusion of a wider array of biomarkers related to β-cell dedifferentiation, particularly epigenetic markers and related signaling pathways. The addition of these biomarkers would enable a deeper exploration of the molecular mechanisms underlying β-cell dysfunction and dedifferentiation in response to the treatments.

We also suggest analyzing the intracellular pathways activated by metformin, semaglutide, and tirzepatide, such as AMPK, mTOR, and ERK, to identify potential overlaps that may limit synergistic effects in apoptosis mitigation. This analysis could provide valuable insights into the molecular mechanisms underlying the observed interactions between these drugs.

Finally, while our study examined cell cycle dynamics, future research should incorporate additional cell-cycle-related biomarkers to further validate the results of the cell cycle assays. This could include the assessment of key regulatory proteins involved in cell cycle progression and checkpoint control, such as cyclins, cyclin-dependent kinases (CDKs), and tumor suppressors. A more comprehensive analysis of these markers would provide additional evidence to support the observed effects on cell cycle regulation and help elucidate the broader implications of the treatments on cellular proliferation and differentiation.

## Figures and Tables

**Figure 1 ijms-26-00421-f001:**
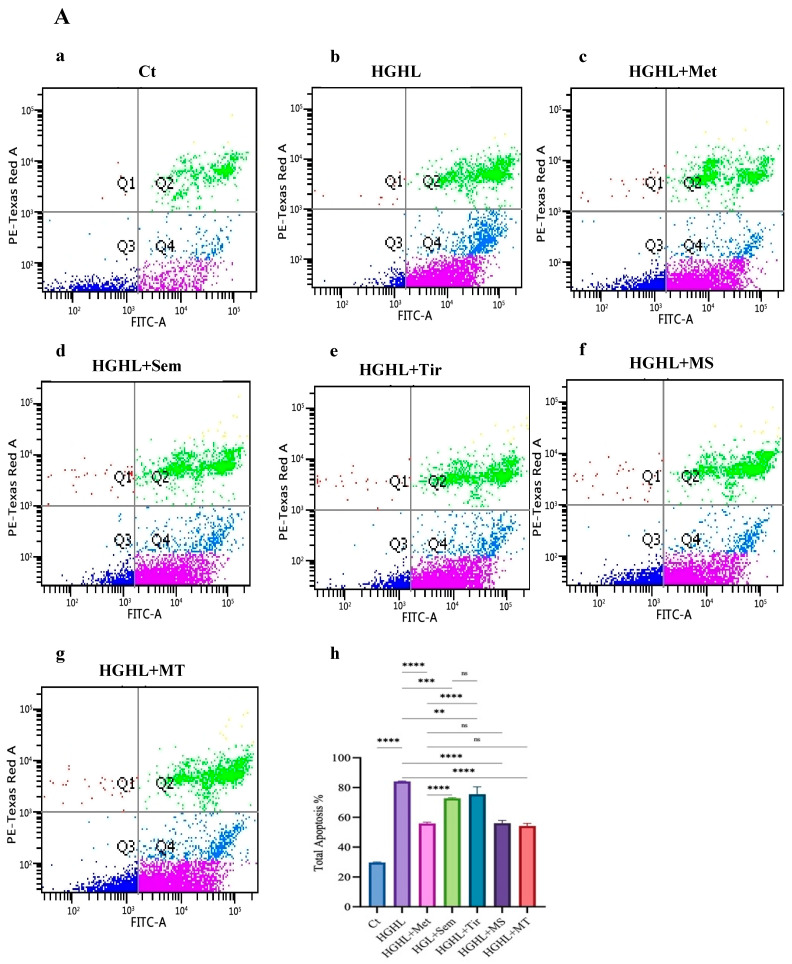
The impacts of 1 mM metformin, 10 nM semaglutide, 10 nM tirzepatide, and their combination on HG-HL-induced apoptosis and levels of individual apoptotic biomarkers. INS-1 β-cells were pretreated with the indicated treatments and then exposed to HG-HL conditions for 48 h, followed by protein extraction, Western blot analysis, or apoptosis assay. (**A**(**a**–**g**)): Flow cytometric graphs showing the four quadrants (Q1–Q4) of the apoptosis assay. (**A**(**h**)): Analysis of the apoptosis assay in HG-HL-induced INS-1 β-cells treated with 1 mM metformin, 10 nM semaglutide, 10 nM tirzepatide, MS, and MT. (**B**(**a**–**h**)): Western blot analysis of apoptotic biomarkers, normalized against β-Actin, in HG-HL-induced INS-1 β-cells treated with 1 mM metformin, 10 nM semaglutide, 10 nM tirzepatide, MS, and MT. (**B**(**i**)): Western blot images of apoptotic biomarkers. Abbreviations: Ct (control), Met (metformin), Sem (semaglutide), Tir (tirzepatide), MS (metformin + semaglutide), MT (metformin + tirzepatide), and HG-HL (high glucose + high lipid). All data are presented as mean ± SD, *n* = 3 measurements. Asterisks indicate significant differences: * *p* < 0.05; ** *p* < 0.01; *** *p* < 0.001; **** *p* < 0.0001; ns—non-significant.

**Figure 2 ijms-26-00421-f002:**
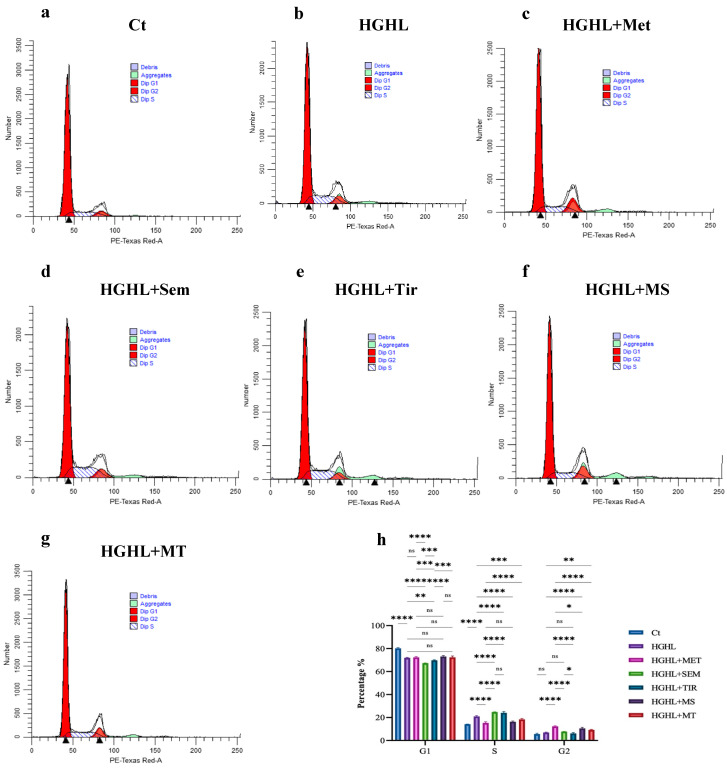
Cell cycle assay of HG-HL-induced INS-1 β-cells treated with 1 mM metformin, 10 nM semaglutide, 10 nM tirzepatide, MS, and MT, analyzed by flow cytometry. Treated cells were fixed with 70% ethanol, washed with PBS, treated with RNase, and stained with propidium iodide (PI) for the assay. (**a**–**g**): Representative histograms show the distribution of cells in the G1, S, and G2 phases under HG-HL conditions and following treatment with 1 mM metformin, 10 nM semaglutide, 10 nM tirzepatide, MS, and MT. Peaks correspond to DNA content in different cell cycle phases, with shifts indicating changes in cell cycle progression. (**h**): Bar graphs represent the percentage of cells in the G1, S, and G2 phases under HG-HL conditions and after treatment with 1 mM metformin, 10 nM semaglutide, 10 nM tirzepatide, MS, and MT. Abbreviations: Ct (control), Met (metformin), Sem (semaglutide), Tir (tirzepatide), MS (metformin + semaglutide), MT (metformin + tirzepatide), and HG-HL (high glucose + high lipid). Data are presented as mean ± SD (*n* = 3 measurements), with asterisks indicating statistically significant differences (* *p* < 0.05, ** *p* < 0.01, *** *p* < 0.001, **** *p* < 0.0001; ns—non-significant).

**Figure 3 ijms-26-00421-f003:**
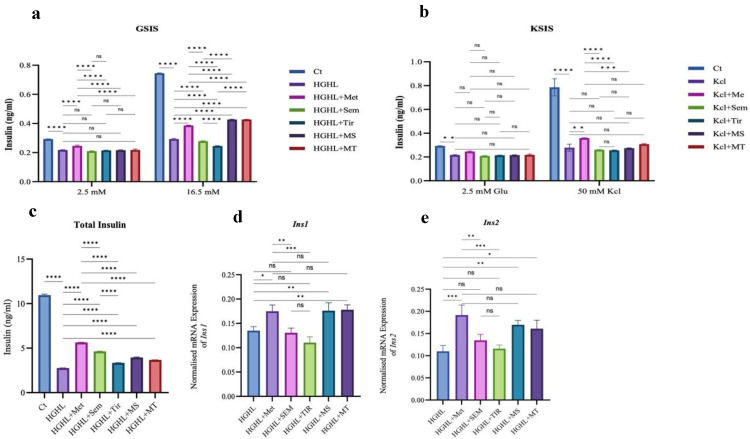
The effects of specific doses of metformin, semaglutide, tirzepatide, MS, and MT on GSIS, KSIS, insulin content, and *Ins* gene expression in HG-HL-challenged β-cells. (**a**,**b**) depict GSIS and KSIS responses of HG-HL-induced β-cells to the treatments in media containing 2.5 mM glucose, 16.5 mM glucose, or 50 mM KCl. (**c**) shows the total insulin content of HG-HL-induced INS-1 β-cells following treatments. (**d**,**e**) illustrate normalized *Ins* gene expression in HG-HL-induced INS-1 β-cells after treatment. Abbreviations: Ct (control), Met (metformin), Sem (semaglutide), Tir (tirzepatide), MS (metformin + semaglutide), MT (metformin + tirzepatide), and HG-HL (high glucose + high lipid). Data are presented as mean ± SD (*n* = 3 measurements), with asterisks indicating statistically significant differences (* *p* < 0.05, ** *p* < 0.01, *** *p* < 0.001, **** *p* < 0.0001; ns—non-significant).

**Figure 4 ijms-26-00421-f004:**
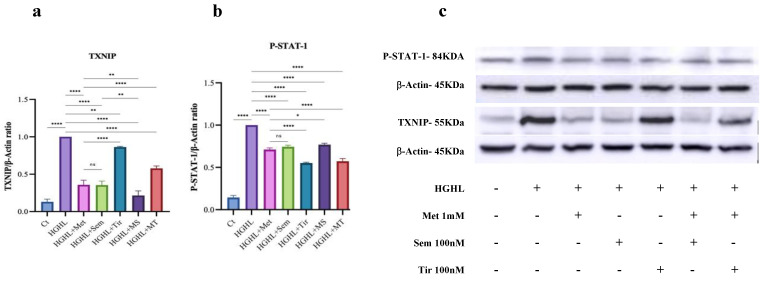
Western blot analysis of P-STAT1 and TXNIP in HG-HL-induced INS-1 β-cells following treatment with 10 mM metformin, 10 nM semaglutide, 10 nM tirzepatide, MS, and MT. INS-1 β-cells were pretreated with specific doses of 1 mM metformin, 10 nM semaglutide, 10 nM tirzepatide, MS, and MT for 2 h and then exposed to HG-HL conditions for 48 h. Protein extraction was performed, and Western blot analysis was conducted. Band intensities were quantified using ImageJ and normalized against β-Actin. (**a**) Western blot analysis of TXNIP. (**b**) Western blot analysis of P-STAT-1. (**c**) Western blot images of TXNIP and P-STAT-1, along with their corresponding housekeeping proteins. Abbreviations: Ct (control), Met (metformin), Sem (semaglutide), Tir (tirzepatide), MS (metformin + semaglutide), MT (metformin + tirzepatide), and HG-HL (high glucose + high lipid). Data are presented as mean ± SD (*n* = 3 measurements), with asterisks indicating statistically significant differences (* *p* < 0.05, ** *p* < 0.01, **** *p* < 0.0001; ns—non-significant).

**Figure 5 ijms-26-00421-f005:**
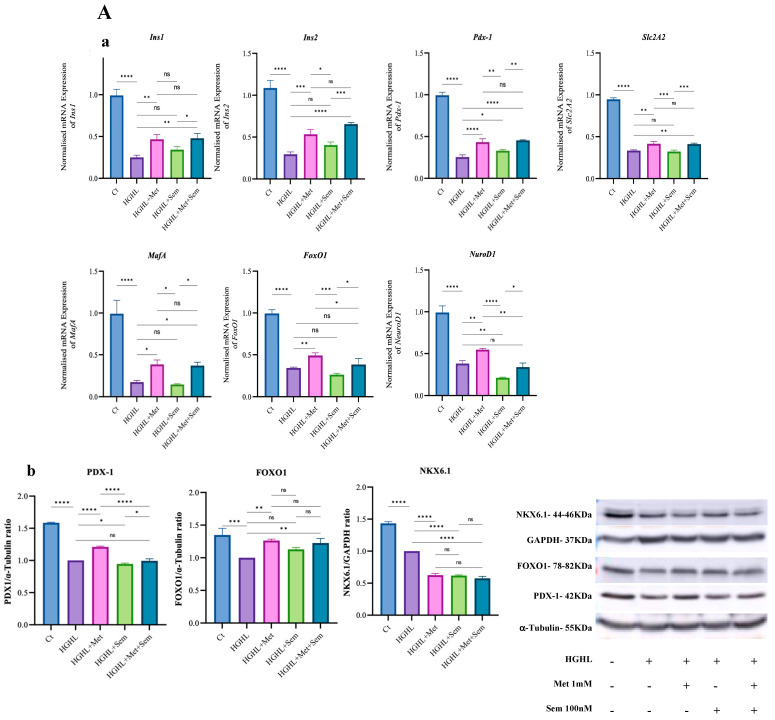
Western blot and qRT-PCR analyses of genes and proteins associated with β-cell identity maintenance in co- and posttreatment experiments. Gene expression analysis of *Ins1*, *Ins2*, *Pdx-1*, *Slc2A2*, *MafA*, *FoxO1*, and *NeuroD1* in HG-HL-induced INS-1 β-cells was performed following co-treatment (**A**(**a**)) or 48 h of post-treatment (**B**(**a**)) with 1 mM metformin, 10 nM semaglutide, and MS. Western blot analysis of PDX-1, FOXO1, and NKX6.1 was conducted under the same conditions for both co-treatment (**A**(**b**)) and post-treatment (**B**(**b**)). Abbreviations: Ct (control), Met (metformin), Sem (semaglutide), Tir (tirzepatide), MS (metformin + semaglutide), MT (metformin + tirzepatide), HG-HL (high glucose + high lipid), and HG-HL+Med (HG-HL+medium). Data are presented as mean ± SD (*n* = 3), with asterisks indicating statistically significant differences (* *p* < 0.05, ** *p* < 0.01, *** *p* < 0.001, **** *p* < 0.0001; ns—non-significant).

## Data Availability

Data are available upon request. Original Western blots were uploaded to the journal.

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
