# Peer review of "Single and Combined Impact of Semaglutide, Tirzepatide, and Metformin on β-Cell Maintenance and Function Under High-Glucose–High-Lipid Conditions: A Comparative Study"

_ijms, 2025, doi:10.3390/ijms26010421_

Round 1
Reviewer 1 Report
Comments and Suggestions for Authors
The manuscript titled Single and Combined Impact of Semaglutide, Tirzepatide, and Metformin on Beta-Cell Maintenance and Function Under High-Glucose-High-Lipid Conditions: A Comparative Study is novel, well-structured, and well-written. I have no further comments for the authors.
Author Response
Thank you for your kind words.
Reviewer 2 Report
Comments and Suggestions for Authors
In the manuscript entitled “Single and Combined Impact of Semaglutide, Tirzepatide, and 2 Metformin on b-Cells Maintenance and Function Under High- 3 Glucose-High-Lipid Conditions: A Comparative Study” the authors show the comparative effects of three drugs—semaglutide, tirzepatide, and metformin—on the maintenance and function of INS-1 β cells under high-glucose, high-lipid (HG-HL) conditions, typical of type 2 diabetes (T2D). The combination of metformin with semaglutide or tirzepatide was shown to be more effective in improving β-cell function and survival under conditions of metabolic stress typical of T2D. These results suggest the therapeutic potential of drug combinations in the treatment of diabetes.
The work is well written and the experiments are well thought out and well done.
Minor points
1. The use of a single concentration for semaglutide and tirzepatide (10 nM) limits the understanding of their efficacy. The observed effects may not reflect the full modulatory potential of the drug. Therefore, a possible suggestion for the authors is to include a dose-response analysis for 10 nM semaglutide and tirzepatide. Examine lower and higher doses to assess any additive or synergistic effects with metformin.
2. The experiments only compare co-treatment and post-treatment treatments. The optimal timing of drug effects is not explored. I recommend the authors to perform experiments with different time intervals to monitor how apoptotic markers and gene expression vary over time
3. Furthermore, the evaluation is limited to a specific set of apoptotic and metabolic biomarkers. It is necessary to include markers of oxidative stress and inflammation, such as ROS, NF-κB or IL-6, to better understand the mechanism of action.
4. The results shown by the authors indicate that some combinations do not completely restore Pdx1, Foxo1 and NeuroD1. Gene silencing or overexpression experiments of these transcription factors should be performed to determine their critical role in mitigating apoptosis and regulating insulin secretion.
5. Another aspect to consider is that the combination of metformin with semaglutide or tirzepatide did not show synergistic effects on the mitigation of apoptosis. I therefore suggest analyzing the intracellular pathways activated by each drug (e.g. AMPK, mTOR, ERK) to determine if there are overlaps that limit synergy.
Author Response
Minor points
- The use of a single concentration for semaglutide and tirzepatide (10 nM) limits the understanding of their efficacy. The observed effects may not reflect the full modulatory potential of the drug. Therefore, a possible suggestion for the authors is to include a dose-response analysis for 10 nM semaglutide and tirzepatide. Examine lower and higher doses to assess any additive or synergistic effects with metformin.
Answer: We appreciate the valuable suggestion regarding the inclusion of a dose-response analysis. We acknowledge that using a single concentration (10 nM) of semaglutide and tirzepatide might limit a full understanding of their modulatory potential. However, our decision to focus on a single dose was made to avoid adding significant complexity to the study. For instance, including three doses of semaglutide and tirzepatide, along with three combinations of these two substances with one dose of metformin, would result in 13 treatment groups. When combined with a negative control and a positive control, this would increase the total to 15 treatments, complicating both the assays and subsequent comparisons.
It is important to note that we selected the single dose of each treatment based on two factors: (1) alignment with findings from previous studies, as detailed in the paper. (2) The concentration of 10 nM for both semaglutide and tirzepatide was chosen because it aligns well with the pharmacologically relevant doses of these medications in humans.
Weekly doses of 0.5–1 mg semaglutide produce steady-state plasma concentrations of approximately 10–20 nM, while weekly doses of 5–15 mg tirzepatide result in plasma concentrations of about 15–25 nM. These therapeutic plasma concentrations closely match the 10 nM concentration used in our study, supporting the relevance and translational potential of our chosen doses.
- The experiments only compare co-treatment and post-treatment treatments. The optimal timing of drug effects is not explored. I recommend the authors to perform experiments with different time intervals to monitor how apoptotic markers and gene expression vary over time.
Answer: We appreciate the valuable suggestion regarding exploring different time intervals to evaluate the timing of drug effects. While we agree that investigating the temporal dynamics of drug treatments could provide valuable insights into the optimal timing of therapeutic interventions, our prior observations and pilot studies guided the decision to focus on the 48-hour time point.
Specifically, we found that extending the incubation with HGHL beyond 48 hours significantly reduced cell viability and function 1,2, leaving insufficient functional cells to perform assays effectively. Conversely, a 24-hour incubation with HGHL did not induce measurable apoptosis, limiting the ability to assess treatment effects.
In addition to these, the selection of the 48-hour time point was further supported by prior studies in the literature 3,4, which identified this duration as optimal for inducing measurable apoptosis while maintaining cell viability for downstream analyses.
We believe this approach balances experimental feasibility with the ability to capture relevant cellular responses, but we recognize the potential value of exploring additional time points in future studies.
- Furthermore, the evaluation is limited to a specific set of apoptotic and metabolic biomarkers. It is necessary to include markers of oxidative stress and inflammation, such as ROS, NF-κB or IL-6, to better understand the mechanism of action.
Answer: We appreciate the insightful suggestion to include additional markers of oxidative stress and inflammation, such as ROS, NF-κB, or IL-6, to better understand the mechanism of action. We agree that these markers could provide valuable information about the broader biological context of the drug treatments.
However, our study focused on a specific set of apoptotic and metabolic biomarkers to maintain a clear and manageable scope for the current investigation. Moreover, based on previous studies, the primary source of inflammation and inflammatory responses in the pancreas is the infiltrating macrophages 5-9. As a result, we have addressed this aspect in a separate project, where we investigated the impact of these medications on THP-1 macrophages.
- The results shown by the authors indicate that some combinations do not completely restore Pdx1, Foxo1 and NeuroD1. Gene silencing or overexpression experiments of these transcription factors should be performed to determine their critical role in mitigating apoptosis and regulating insulin secretion.
Answer: We appreciate the suggestion to conduct gene silencing or overexpression experiments of Pdx1, Foxo1, and NeuroD1 to further elucidate their role in mitigating apoptosis and regulating insulin secretion. We agree that these transcription factors may play a critical role in the observed effects, and investigating their specific contributions through such experiments would provide valuable insights into their mechanistic involvement.
However, our current study focused on assessing the effects of the treatments on a set of well -known beta cells-enriched biomarkers and transcription factors like Pdx1, Foxo1, and NeuroD1. Accordingly, the role of these transcription factors on the function and maintenance of the beta cells are well-established 10-17.
- Another aspect to consider is that the combination of metformin with semaglutide or tirzepatide did not show synergistic effects on the mitigation of apoptosis. I therefore suggest analyzing the intracellular pathways activated by each drug (e.g. AMPK, mTOR, ERK) to determine if there are overlaps that limit synergy.
Answer: We appreciate the thoughtful suggestion to analyze the intracellular pathways activated by metformin, semaglutide, and tirzepatide, such as AMPK, mTOR, and ERK, to determine potential overlaps that could limit synergistic effects. This is an important consideration, and understanding the specific molecular mechanisms underlying the observed lack of synergy could provide valuable insights into the interactions between these drugs.
While we did not include an in-depth analysis of these pathways in the current study, we recognize that the potential for pathway cross-talk could influence the therapeutic effects of these combinations. In fact, metformin is known to primarily activate AMPK, whereas semaglutide and tirzepatide may involve different pathways, such as those related to GLP-1 receptor signaling. As you suggested, if there are overlapping pathways, it could explain the absence of synergy in apoptosis mitigation.
We added a suggestion in this regard to the future direction of the paper.
References
1 Ghasemi Gojani, E., Wang, B., Li, D., Kovalchuk, O. & Kovalchuk, I. The Effects of Psilocybin on Lipopolysaccharide-Induced Inflammation in THP-1 Human Macrophages. Psychoactives 3, 48-64 (2024).
2 Gojani, E. G., Wang, B., Li, D.-P., Kovalchuk, O. & Kovalchuk, I. The Impact of Psilocybin on High Glucose/Lipid-Induced Changes in INS-1 Cell Viability and Dedifferentiation. Genes 15, 183 (2024).
3 Wang, Y. et al. Telmisartan protects against high glucose/high lipid‐induced apoptosis and insulin secretion by reducing the oxidative and ER stress. Cell biochemistry and function 37, 161-168 (2019).
4 Zhang, D., Xie, T. & Leung, P. S. Irisin ameliorates glucolipotoxicity-associated β-cell dysfunction and apoptosis via AMPK signaling and anti-inflammatory actions. Cellular Physiology and Biochemistry 51, 924-937 (2018).
5 Jourdan, T. et al. Activation of the Nlrp3 inflammasome in infiltrating macrophages by endocannabinoids mediates beta cell loss in type 2 diabetes. Nature medicine 19, 1132-1140 (2013).
6 Carrero, J. A. et al. Resident macrophages of pancreatic islets have a seminal role in the initiation of autoimmune diabetes of NOD mice. Proceedings of the National Academy of Sciences 114, E10418-E10427 (2017).
7 Ying, W., Fu, W., Lee, Y. S. & Olefsky, J. M. The role of macrophages in obesity-associated islet inflammation and β-cell abnormalities. Nature Reviews Endocrinology 16, 81-90 (2020).
8 Nackiewicz, D. et al. TLR2/6 and TLR4-activated macrophages contribute to islet inflammation and impair beta cell insulin gene expression via IL-1 and IL-6. Diabetologia 57, 1645-1654 (2014).
9 Nordmann, T. M. et al. The role of inflammation in β-cell dedifferentiation. Scientific reports 7, 6285 (2017).
10 Zhang, Y. et al. PDX-1: a promising therapeutic target to reverse diabetes. Biomolecules 12, 1785 (2022).
11 Gao, T. et al. Pdx1 maintains β cell identity and function by repressing an α cell program. Cell metabolism 19, 259-271 (2014).
12 Kobayashi, M. et al. FoxO1 as a double-edged sword in the pancreas: analysis of pancreas-and β-cell-specific FoxO1 knockout mice. American Journal of Physiology-Endocrinology and Metabolism 302, E603-E613 (2012).
13 Zhang, T. et al. FoxO1 plays an important role in regulating β-cell compensation for insulin resistance in male mice. Endocrinology 157, 1055-1070 (2016).
14 Szydłowski, M., Jabłońska, E. & Juszczyński, P. FOXO1 transcription factor: a critical effector of the PI3K-AKT axis in B-cell development. International reviews of immunology 33, 146-157 (2014).
15 Romer, A. I., Singer, R. A., Sui, L., Egli, D. & Sussel, L. Murine perinatal β-cell proliferation and the differentiation of human stem cell–derived insulin-expressing cells require NEUROD1. Diabetes 68, 2259-2271 (2019).
16 Ono, Y. & Kataoka, K. MafA, NeuroD1, and HNF1β synergistically activate the Slc2a2 (Glut2) gene in β-cells. Journal of molecular endocrinology 67, 71-82 (2021).
17 Nlend, R. N. et al. Cx36 is a target of Beta2/NeuroD1, which associates with prenatal differentiation of insulin-producing β cells. The Journal of membrane biology 245, 263-273 (2012).
18 Killion, E. A. et al. Glucose-dependent insulinotropic polypeptide receptor therapies for the treatment of obesity, do agonists= antagonists? Endocrine reviews 41, 1-21 (2020).
19 Killion, E. A. et al. Chronic glucose-dependent insulinotropic polypeptide receptor (GIPR) agonism desensitizes adipocyte GIPR activity mimicking functional GIPR antagonism. Nature communications 11, 4981 (2020).
20 Killion, E. A. et al. Anti-obesity effects of GIPR antagonists alone and in combination with GLP-1R agonists in preclinical models. Science Translational Medicine 10, eaat3392 (2018).
21 Campbell, J. E. Targeting the GIPR for obesity: To agonize or antagonize? Potential mechanisms. Molecular metabolism 46, 101139 (2021).
22 Lu, S.-C. et al. GIPR antagonist antibodies conjugated to GLP-1 peptide are bispecific molecules that decrease weight in obese mice and monkeys. Cell Reports Medicine 2 (2021).
Reviewer 3 Report
Comments and Suggestions for Authors
Dear Authors
thank you for submitting your manuscript entitled (Single and Combined Impact of Semaglutide, Tirzepatide, and Metformin on -Cells Maintenance and Function Under High-Glucose-High-Lipid Conditions: A Comparative Study) for consideration in IJMS
the manuscript discuss the impact of use of GLP-1R agonist Semaglutide or Tirzepatide with or without metformin on INS-1 insulinoma cell line stimulated by HGHL conditions
The abstract contain GSIS abbreviation without being mentioned before
The study design, involving five groups (M, S, T, MS, MT) along with positive and negative controls, appears somewhat unclear. This is because the comparison between groups (at least 10 comparisons per figure sometimes 15 comparison) relies on determining whether there is any synergistic or additive effect.
Figure 5 does not include all the groups
western blot data of T group alone is the most interesting and clear data however it receive the least discussion
Please reduce using self-citation whenever possible
English need to be revised as a lot of typos were detected
Overall the study looks promising but needs further revision
Author Response
- The abstract contain GSIS abbreviation without being mentioned before.
Answer: Thank you for the comment. It has been corrected.
The study design, involving five groups (M, S, T, MS, MT) along with positive and negative controls, appears somewhat unclear. This is because the comparison between groups (at least 10 comparisons per figure sometimes 15 comparison) relies on determining whether there is any synergistic or additive effect.
Answer: We appreciate the valuable feedback regarding the study design and the complexity of comparisons. We would like to clarify that the comparisons between the groups were made only where there were significant differences, allowing us to extract more meaningful results and ensure a comprehensive analysis. This approach was intended to provide a detailed understanding of the interactions between the treatments.
We understand that the number of comparisons (at least 10 per figure, sometimes 15) might seem overwhelming, but we focused on comparing groups where there was a reasonable expectation of synergistic or additive effects. If the complexity of these comparisons is making the interpretation of any specific results more difficult, we are open to addressing this concern. We could refine the analysis by providing clearer distinctions between specific comparisons, or adjusting the presentation of the data to make it more straightforward.
- Figure 5 does not include all the groups
Answer: Thank you for the comment. That part of the study was based on the results from the apoptosis and cell cycle analyses. Our findings with 10 nM semaglutide and the combination of metformin and semaglutide (MS) on HG-HL-induced apoptosis and S-phase entry indicated that this combination was more effective than the T and MT. Therefore, we decided to conduct this part of the experiment using only 1 mM metformin, 10 nM semaglutide, and the MS combination.
- Western blot data of T group alone is the most interesting and clear data however it receives the least discussion
Answer: We appreciate your suggestion regarding the Western blot data for the tirzepatide group. We agree that this data is particularly intriguing and warrants further discussion. As noted in the manuscript, tirzepatide, unlike metformin or semaglutide, increased the levels of apoptotic biomarkers such as C-Caspase-3, C-Caspase-9, and Bim EL under HG-HL conditions. This aligns with the apoptosis assay results and the behavior of TXNIP protein, suggesting a unique and complex mechanism of action for tirzepatide.
We intentionally kept the discussion brief to maintain a focused scope, as the primary aim of the study was to evaluate the comparative effects of the treatments. However, based on this feedback, we expanded the discussion to explore the implications of tirzepatide’s unique effects.
Additionally, we added the following explanation to the discussion part “This observation may be linked to the prolonged exposure to GIP or GIPR agonists, which has been shown to lead to receptor desensitization, reducing their effectiveness. Studies indicate that extended GIPR activation can result in receptor desensitization, potentially explaining the stimulatory effects of tirzepatide on some apoptotic biomarkers 18-20. In this regard, it has been observed that, similar to the remarkable therapeutic effectiveness of dual GIPR/GLP-1R agonists, the combination of GLP-1R agonism with GIPR antagonism also demonstrates significant therapeutic benefits 20-22. These findings underscore the complexity of GIPR mechanisms of action and the synergistic interplay between GIPR and GLP-1R, emphasizing the need for further research to fully elucidate their dual metabolic and apoptotic effects.”
- Please reduce using self-citation whenever possible
Answer: We understand the importance of maintaining a balanced and relevant citation list. In this study, self-citations were included only to reference prior work that directly supports the methodology, findings, or interpretation of results. However, we have carefully reviewed the manuscript and replaced or removed self-citations wherever alternative references were equally appropriate and relevant. Specifically, references 4 and 7 have been replaced with equally suitable alternatives.
- English need to be revised as a lot of typos were detected
Answer: We corrected typos where we found them.
References
1 Ghasemi Gojani, E., Wang, B., Li, D., Kovalchuk, O. & Kovalchuk, I. The Effects of Psilocybin on Lipopolysaccharide-Induced Inflammation in THP-1 Human Macrophages. Psychoactives 3, 48-64 (2024).
2 Gojani, E. G., Wang, B., Li, D.-P., Kovalchuk, O. & Kovalchuk, I. The Impact of Psilocybin on High Glucose/Lipid-Induced Changes in INS-1 Cell Viability and Dedifferentiation. Genes 15, 183 (2024).
3 Wang, Y. et al. Telmisartan protects against high glucose/high lipid‐induced apoptosis and insulin secretion by reducing the oxidative and ER stress. Cell biochemistry and function 37, 161-168 (2019).
4 Zhang, D., Xie, T. & Leung, P. S. Irisin ameliorates glucolipotoxicity-associated β-cell dysfunction and apoptosis via AMPK signaling and anti-inflammatory actions. Cellular Physiology and Biochemistry 51, 924-937 (2018).
5 Jourdan, T. et al. Activation of the Nlrp3 inflammasome in infiltrating macrophages by endocannabinoids mediates beta cell loss in type 2 diabetes. Nature medicine 19, 1132-1140 (2013).
6 Carrero, J. A. et al. Resident macrophages of pancreatic islets have a seminal role in the initiation of autoimmune diabetes of NOD mice. Proceedings of the National Academy of Sciences 114, E10418-E10427 (2017).
7 Ying, W., Fu, W., Lee, Y. S. & Olefsky, J. M. The role of macrophages in obesity-associated islet inflammation and β-cell abnormalities. Nature Reviews Endocrinology 16, 81-90 (2020).
8 Nackiewicz, D. et al. TLR2/6 and TLR4-activated macrophages contribute to islet inflammation and impair beta cell insulin gene expression via IL-1 and IL-6. Diabetologia 57, 1645-1654 (2014).
9 Nordmann, T. M. et al. The role of inflammation in β-cell dedifferentiation. Scientific reports 7, 6285 (2017).
10 Zhang, Y. et al. PDX-1: a promising therapeutic target to reverse diabetes. Biomolecules 12, 1785 (2022).
11 Gao, T. et al. Pdx1 maintains β cell identity and function by repressing an α cell program. Cell metabolism 19, 259-271 (2014).
12 Kobayashi, M. et al. FoxO1 as a double-edged sword in the pancreas: analysis of pancreas-and β-cell-specific FoxO1 knockout mice. American Journal of Physiology-Endocrinology and Metabolism 302, E603-E613 (2012).
13 Zhang, T. et al. FoxO1 plays an important role in regulating β-cell compensation for insulin resistance in male mice. Endocrinology 157, 1055-1070 (2016).
14 Szydłowski, M., Jabłońska, E. & Juszczyński, P. FOXO1 transcription factor: a critical effector of the PI3K-AKT axis in B-cell development. International reviews of immunology 33, 146-157 (2014).
15 Romer, A. I., Singer, R. A., Sui, L., Egli, D. & Sussel, L. Murine perinatal β-cell proliferation and the differentiation of human stem cell–derived insulin-expressing cells require NEUROD1. Diabetes 68, 2259-2271 (2019).
16 Ono, Y. & Kataoka, K. MafA, NeuroD1, and HNF1β synergistically activate the Slc2a2 (Glut2) gene in β-cells. Journal of molecular endocrinology 67, 71-82 (2021).
17 Nlend, R. N. et al. Cx36 is a target of Beta2/NeuroD1, which associates with prenatal differentiation of insulin-producing β cells. The Journal of membrane biology 245, 263-273 (2012).
18 Killion, E. A. et al. Glucose-dependent insulinotropic polypeptide receptor therapies for the treatment of obesity, do agonists= antagonists? Endocrine reviews 41, 1-21 (2020).
19 Killion, E. A. et al. Chronic glucose-dependent insulinotropic polypeptide receptor (GIPR) agonism desensitizes adipocyte GIPR activity mimicking functional GIPR antagonism. Nature communications 11, 4981 (2020).
20 Killion, E. A. et al. Anti-obesity effects of GIPR antagonists alone and in combination with GLP-1R agonists in preclinical models. Science Translational Medicine 10, eaat3392 (2018).
21 Campbell, J. E. Targeting the GIPR for obesity: To agonize or antagonize? Potential mechanisms. Molecular metabolism 46, 101139 (2021).
22 Lu, S.-C. et al. GIPR antagonist antibodies conjugated to GLP-1 peptide are bispecific molecules that decrease weight in obese mice and monkeys. Cell Reports Medicine 2 (2021).
Round 2
Reviewer 3 Report
Comments and Suggestions for Authors
Thank you for addressing the comments